# Understanding Multi-Task Scaling in Machine Translation

## Abstract

In this work, we provide a large-scale empirical study of the scaling properties of multilingual (multitask) neural machine translation models. We examine how increases in the model size affect the model performance and investigate the role of the individual task weights on the scaling behavior. We find that these weights only affect the multiplicative factor of the scaling law and in particular, the scaling exponent is unaffected by them. Through a novel joint scaling law formulation, we compute the *effective number of parameters* allocated to each task and examine the role of language similarity in the scaling behavior of our models. We find minimal evidence that language similarity has any impact. In contrast, "direction" of the multilinguality plays a significant role, with models translating from multiple languages *into* English having a larger number of effective parameters per task than their reversed counterparts. Finally, we leverage our observations to predict the performance of multilingual models trained with *any* language weighting at *any* scale, greatly reducing efforts required for task balancing in large multitask models. Our findings apply to both in-domain and out-of-domain test sets and to multiple evaluation metrics, such as ChrF and BLEURT.

## 1 Introduction

Over the past few years, scaling has emerged as a popular and effective way to improve the performance of neural networks (Brown et al., 2020; Chowdhery et al., 2022; Lepikhin et al., 2020). Given the costs associated with training large state-of-the-art neural models, much work has gone into understanding their scaling properties and predicting the evolution of their performance with scale through **scaling laws**. Such scaling laws have been instrumental in guiding the model development efforts across a variety of domains such as computer vision (Zhai et al., 2022), language modelling (Kaplan et al., 2020; Hoffmann et al., 2022), and neural machine translation (Ghorbani et al., 2022).

Despite these impressive developments, as of yet, most of the scaling laws studies available in the literature only focus on single-task models. On the contrary, current massive neural models are often trained to solve more than one task across one or more modalities (Chowdhery et al., 2022; Sanh et al., 2022; Reed et al., 2022). This disconnect from the current research frontier limits the applicability of the scaling laws in guiding model development decisions. In particular, currently available scaling laws studies are unable to inform the decision process on how to **balance the different tasks effectively** at training time. Without such guidance, practitioners often have to rely on cumbersome and costly approaches such as approximate grid search to inform their decision-making. Such approaches quickly become infeasible as the problem scale grows.

In this paper, we take the initial step towards developing a quantitative understanding of the scaling behavior for multitask models. We choose multilingual neural machine translation (MNMT) as the setup for this initial study. This choice is motivated by several reasons: MNMT provides a popular setup with mature benchmarks and substantial literature on scaling (Lepikhin et al., 2020; Costa-jussà et al., 2022; Bapna et al., 2022; Huang et al., 2019). Moreover, recent results on scaling laws for single-task MT models provide a natural starting point for our study (Ghorbani et al., 2022; Bansal et al., 2022; Gordon et al., 2021; Zhang et al., 2022). Finally, recent findings on the optimization dynamics of MNMT models greatly simplify our study by removing the need to examine the role of the optimization algorithm in our results (Xin et al., 2022).

For our analysis, we train over 200 MNMT models (ranging from 20M to 1B non-embedding parameters) and systematically examine their scaling behaviors. We focus our investigation on the **data rich-compute rich regime** where we have access to vast amounts of training data for all the tasks (i.e. language pairs)[1] and the model is trained to near convergence. Here, the main bottleneck in the model performance is due to the lack of model capacity. We establish the following observations:

- For each fixed task $i$ and task weighting $\boldsymbol{w}$, the evolution of the test cross-entropy loss ($\mathcal{L}$) with model size ($N$) follows a scaling law that resembles the scaling behavior of single-task models:

$$\mathcal{L}_i(N; \boldsymbol{w}) \approx \beta_{\boldsymbol{w},i} N^{-\alpha_{\boldsymbol{w},i}} + L_\infty^{(\boldsymbol{w},i)}. \tag{1}$$

  Furthermore, we find that changes in the task weightings only affect the multiplicative factor $\beta$. The scaling exponent $\alpha$ and the irreducible loss $L_\infty$ are unaffected by these changes. In other words, scaling multi-task models will improve their performance in a task at the same rate independently of its weight on the optimization objective.

- We leverage these findings to propose a scaling law that jointly predicts the performance for all tasks and weightings considered, and use it to examine how the model splits its capacity in between the tasks by computing the **effective number of parameters** allocated to each task (subsection 3.3)

- We examine the popular belief that training multilingual models in similar languages is more effective than training models in unrelated languages. Surprisingly, for the high-resource language pairs considered, we don't observe any significant differences in the scaling behavior of models trained to translate from English into related languages (En→{De, Fr}) with models trained in unrelated languages (En→{De, Zh}). In contrast, we observe that models trained to translate from multiple languages into English (XX→En) benefit much more from multitasking compared to trained on translation out of English (En→XX).

- In Section 3.4, we use simple approximations to $f_i(\boldsymbol{w})$ to provide a scaling law that predicts **the full task performance trade-off frontier** as a function of the model size $N$ (See Figure 7). We describe how these predictions can be utilized for guiding task balancing in the development of massive models.

## 2 BACKGROUND

### 2.1 NEURAL SCALING LAWS

Recent research suggests that the performance of large neural models is well-predicted by a smooth function of the fundamental problem parameters: the model size $N$ [2], the size of the training data $D$, and the amount of compute used for training $C$ (Hestness et al., 2017; Rosenfeld et al., 2019; Kaplan et al., 2020; Hernandez et al., 2021). The most relevant of these studies to ours is Ghorbani et al. (2022) where the authors study the effects of increasing the model size for single-task NMT models in the data-rich ($D \to \infty$), compute-rich ($C \to \infty$) regime. In this setting, the authors show that the following *bivariate* law describes the scaling behavior of encoder-decoder Transformers

$$\mathcal{L}(N_e, N_d) = \beta N_e^{-p_e} N_d^{-p_d} + L_\infty. \tag{2}$$

Here, $N_e$ and $N_d$ correspond to the number of parameters in the encoder and decoder respectively and $L_\infty$ corresponds to the irreducible loss associated with the task. $\{\beta, p_e, p_d, L_\infty\}$ are the parameters of the scaling law that need to be empirically estimated from the data.

In addition, Ghorbani et al. (2022) examine the question of optimally allocating parameters between the encoder and the decoder. They show that in order to observe the optimal scaling behavior, one needs to proportionally scale the encoder and the decoder together. Under such scaling scheme, Equation 2 simplifies to

$$\mathcal{L}(N) = \beta N^{-\alpha} + L_\infty, \tag{3}$$

---

[1]Using machine translation terminology, all language pairs are *high-resource*.

[2]Following the literature conventions, we only consider the non-embedding layers when computing $N$.

which is similar to the scaling behavior observed in other domains such as computer vision (Zhai et al., 2022) and autoregressive generative models (Henighan et al., 2020).

Based on these results, to achieve the optimal scaling behavior, we adopt the proportional encoder-decoder scaling scheme for our experiments. A detailed overview of the size and architecture of our models is presented in Appendix A.

## 2.2 MULTITASK OPTIMIZATION

We focus our investigation on the supervised learning setup where the model parameters $\boldsymbol{\theta} \in \mathbb{R}^p$ are trained on $K$ different tasks simultaneously. In multilingual MT, each task corresponds to translation for a different language pair. We denote the loss associated with task $i$ with $\mathcal{L}_i(\boldsymbol{\theta})$.

Multitask models are often trained by minimizing a convex combination of the per-task losses:

$$\hat{\boldsymbol{\theta}}(\boldsymbol{w}) = \arg\min \sum_{i=1}^{K} \boldsymbol{w}_i \mathcal{L}_i(\boldsymbol{\theta}) \quad \text{where} \quad \boldsymbol{w} > 0, \quad \sum_{i=0}^{K} \boldsymbol{w}_i = 1 \tag{4}$$

Here, $\boldsymbol{w}$ is a fixed vector of the task weights, determined apriori by the practitioner to emphasize her preferences on the balancing of the tasks. This so-called **scalarization** approach is highly popular in the community due to its effectiveness and simplicity.[3] In fact, despite this simplicity, recent results on multitask optimization suggest that scalarization achieves performances on par or better than bespoke optimizers designed specifically for multitask models (Xin et al., 2022; Kurin et al., 2022).

In current large text models, such explicit scalarization is rare. Instead, scalarization is often implemented **implicitly**, by sampling observations from each task proportionally to that task's weight. Proportional sampling produces (in expectation) the same overall loss function as explicit scalarization but with much less engineering complexity.

# 3 EFFECTS OF SCALE IN MULTILINGUAL MT

## 3.1 EXPERIMENTAL SETUP

We use the (pre-LN) encoder-decoder Transformer architecture in our models (Xiong et al., 2020; Vaswani et al., 2017). We train models of up to 8 sizes, approximately ranging from 20M to 1B (non-embedding) parameters. When scaling encoder-decoder Transformers, to achieve the optimal scaling behavior, we scale the encoder and the decoder proportionally by increasing the model dimension and the number of layers in tandem. See Appendix A for a detailed overview.

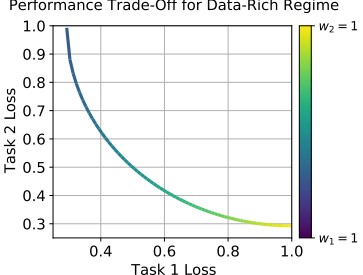

Figure 1: Cartoon representation of the performance trade-off frontier for a hypothetical model.

For our experiments, we train two cohorts of models: En→XX and XX→En. For En→XX cohort, we train multilingual model for translation from English to {German (De), Chinese (Zh)} and {German (De), French (Fr)}. For XX→En cohort, we present results for {De, Zh}→En.

We use the *implicit* scalarization approach to train our models; each observation in the training batch is chosen from the first language pair with probability $p$ and the second language pair with probability $1 - p$.[4] For our experiments, we choose $p$ from the set

$$p \in \{0, 0.05, 0.1, 0.3, 0.5, 0.7, 0.9, 0.95, 1\}. \tag{5}$$

For En→XX models, to avoid confusing the model, we prepend a language token to the source sentence specifying the target language (e.g. `<2de>`). The models are trained using a per-token

---

[3]See (Boyd & Vandenberghe, 2004) for more a detailed discussion of scalarization.

[4]To emphasize the fact that we use sampling-based scalarization, we replace $\boldsymbol{w}$ with $p$ in our notation.

cross-entropy loss and the Adafactor optimizer (Shazeer & Stern, 2018), using a fixed batch size of 500K tokens. To mirror the compute-rich regime as closely as possible, we trained our models to near convergence. In practice, this translates to training our smaller models ($< 500M$ parameters) for 500K gradient steps and our larger models for 1M steps.

To place our models in the data-rich regime, we use a massive in-house web-crawled dataset for training our models. We filter this data using an online data selection procedure (Wang et al., 2018) and high-quality web-domain reference sets, extracting 600M sentences for each language pair. We tokenize this corpus by using a pretrained multilingual SentencePiece Kudo (2018) vocabulary, with a size of 128K sub-words.

We measure the performance of models on both *in-domain* and *out-of-domain* test sets. For the in-domain test set, we extract 2000 sentences from the same in-house datasets used to create the training (ensuring no overlap). For out-of-domain, we use *newstest2019* (Barrault et al., 2019), consisting of 2000 sentence-pairs extracted from aligned news documents.

## 3.2 RESULTS & ANALYSIS

**Understanding Multitask Scaling** We start our analysis by independently examining the model scaling behavior for each individual task weighting $p$ in (5). For each choice of $p$, we fit a scaling law of the form

$$\mathcal{L}_i(N; p) = \beta_{p,i} N^{-\alpha_{p,i}} + L_\infty^{(p,i)} \tag{6}$$

to the empirical (test) performance of models resulting from that task weighting.

Figure 2 presents our findings for En→{De, Zh} models. Each point on the graph corresponds to the empirical test-cross entropy performance of a model at the end of the training.[5] We can see that our per-task-weighting laws are able to capture the scaling behavior of our multilingual models on both language pairs. As expected, when the weight for one of the languages is decreased, the performance of the models on that language decreases for all scales. Our results suggest that the benefits of the increased model size for MNMT models are well-described by a power-law. See Appendix B for similar results for other language pair combinations.

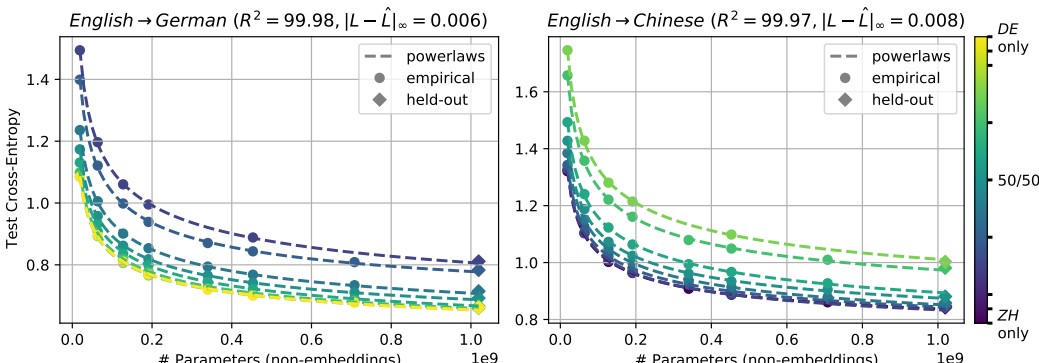

Figure 2: The evolution of the (in-domain) test cross-entropy loss with model size for En→{De, Zh} models, as well as the fitted scaling laws. These scaling laws are **fitted separately for each task weighting**. The color represents the weighting of the languages. The scaling laws are able to capture close to 100% of the variation in the data for both language pairs. Note that we don't show the *zero-shot* behavior.

Figure 4 shows the fitted coefficients of the scaling laws for all $p$. The shaded area marks the one standard deviation uncertainty interval of our estimates.[6] Interestingly, we find that, across all values

---

[5]For low probability tasks, we apply a convergence correction procedure to make up for slow convergence. See the Appendix G for more details.

[6]We gauge the uncertainty in the coefficients by examining the fluctuations in our estimates if our empirical datapoints are perturbed by $\epsilon \overset{\text{i.i.d}}{\sim} \mathcal{N}(0, \sigma^2)$. We choose a conservative $\sigma$ of 1% of the observed empirical loss for each data point.

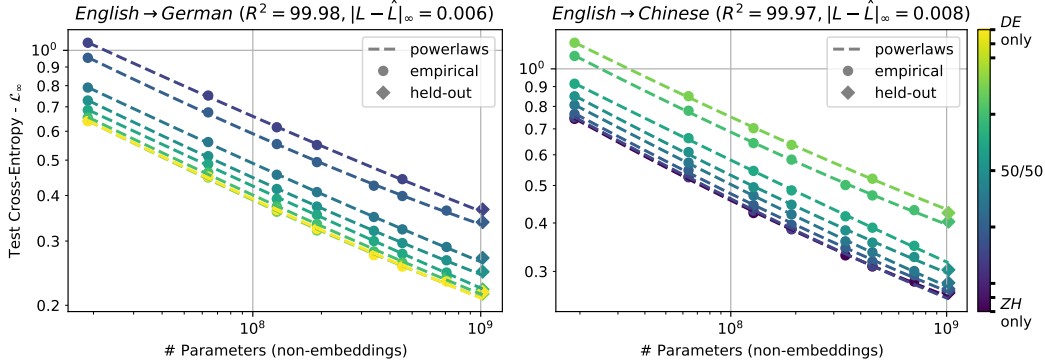

Figure 3: Log-log plot of the evolution of the (in-domain) test cross-entropy loss as we scale. We subtract a constant $L_\infty^{(i)}$, jointly fitted for all the task weights (Equation 7). All lines are nearly parallel, suggesting that the scaling exponent is unchanged for all $p$.

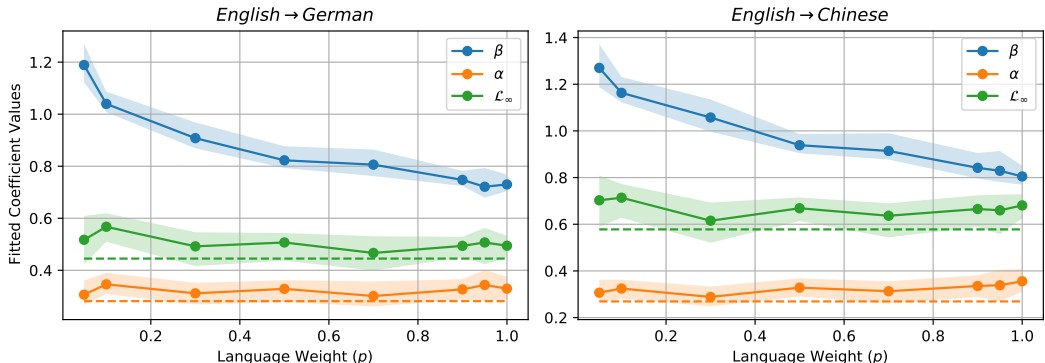

Figure 4: Coefficient values for German (left) and Chinese (right) as a function of the language weight, with the shaded region representing the standard deviation. The dashed lines represent the value of jointly fitted coefficients from Equation 7

of $p$, both the scaling exponent ($\alpha$) and the irreducible loss ($\mathcal{L}_\infty$) seem to be relatively unchanged. In particular, all of our estimated $\alpha$ and $\mathcal{L}_\infty$ parameters are within two standard deviations of each other. In contrast, the multiplicative factor $\beta$ seems to be highly sensitive to the choice of $p$.

Figure 3 visually confirms the assertion that for our models $\alpha_p$ and $L_\infty$ are effectively constant. Here, we have subtracted a fixed constant $L_\infty^{(i)}$ from all the Figure 4 curves corresponding to the task $i$. We then plot results on log-log axes. As the figure suggests, the lines are all near parallel, suggesting that the scaling exponent is unchanged for all $p$. In practical terms this means that, for example, doubling the capacity of a multitask model will reduce its loss by the same $\frac{1}{2^\alpha}$ factor, whether it was trained with 0.1 or 0.9 task weight. This also means that single-task scaling laws can be used to gauge the benefits of scaling multitask models.

**Jointly Modeling Multitask Scaling** Based on the findings above, we make the assumption that the scaling exponents and the irreducible losses are independent of the task weights, and propose a **joint** scaling law of the form

$$\mathcal{L}_i(N; p) \approx \beta_{p,i} N^{-\alpha_i} + L_\infty^{(i)}. \tag{7}$$

Figure 5 shows the fit of this joint scaling law for En→{De, Zh} models evaluated on the in-domain test sets. Note that here, we fit a total of 10 parameters for each task − 8 for $\beta_{p,i}$'s and two for $\alpha_i$ and $L_\infty^{(i)}$. In contrast, in Figure 2, we used 24 overall parameters to capture the scaling behavior for each task. Despite this significant decrease in the number of total fitted parameters, we observe that

our joint laws are able to almost completely capture the scaling behavior. We observe a similar phenomenon for out-of-domain test sets and other language pairs (see Appendix C), further suggesting that the joint law accurately describes the scaling behavior of MNMT models.

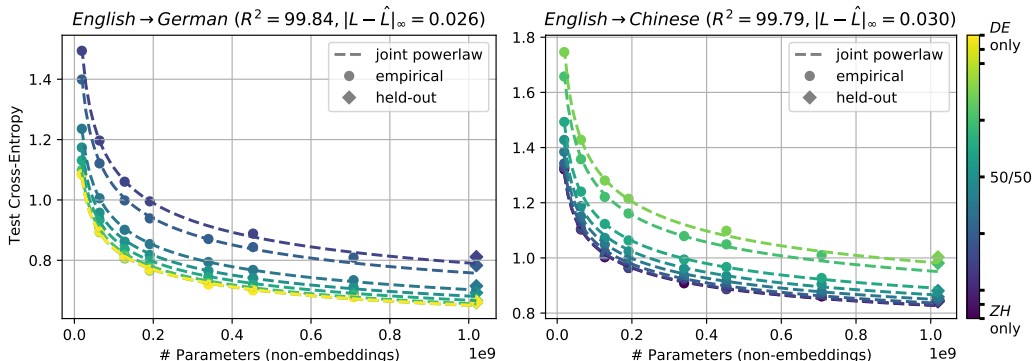

Figure 5: The **joint** scaling law of Equation 7 closely captures the scaling behavior of En→{De, Zh} models. Test loss here is evaluated on in-domain test sets. See Appendix C for similar observations on En→{De, Fr} and {De, Zh}→En models.

## 3.3 EFFECTIVE NETWORK CAPACITY FOR MULTITASK MODELS

We leverage our joint scaling law to examine how MNMT models split their capacity in between the different tasks. We start by defining the notion of **the effective number of parameters**:

**Definition.** *Consider a multitask model in which a task $i$ has been trained with weight $p$. We define the effective number of parameters allocated to $i$, $N_{eff}^{(i,p)}$, to be equal to the number of parameters necessary for a single-task model solely trained on $i$ to reach the same (test loss) performance as the multitask model.*

Mathematically, $N_{\text{eff}}^{(i,p)}$ can be written as the solution of the equation

$$\mathcal{L}_i(N; p) = \mathcal{L}_i(N_{\text{eff}}^{(i,p)}; 1).  \qquad (8)$$

A simple derivation yields that [7]

$$N_{\text{eff}}^{(i,p)} = \left(\frac{\beta_{1,i}}{\beta_{p,i}}\right)^{\frac{1}{\alpha_i}} N.  \qquad (9)$$

Crucially, our calculations suggest that the fraction of parameters allocated to task $i$, which we denote by $f_i(p)$, is independent of the model size:

$$f_i(p) \equiv N_{\text{eff}}^{(i,p)}/N = \left(\frac{\beta_{1,i}}{\beta_{p,i}}\right)^{\frac{1}{\alpha_i}}.  \qquad (10)$$

This observation yields a fundamental, scale-independent quantity that can be leveraged for understanding the interactions between the different tasks in MNMT models.

Figure 6 shows the empirically estimated effective parameter ratios for our models. Several observations are in order:

**Consistency Across Domains:** In Figure 6 (left), we compare the capacity splitting behavior of the models on in-domain and out-of-domain (newstest19) test sets. Even though the scaling laws coefficients for in-domain and out-of-domain test sets differ, we observe that the capacity splitting behavior is mostly unchanged with different test sets. These findings hint at some measure of universality across test domains on how MNMT models divide their capacity and share their parameters.

---

[7]See Appendix D for details.

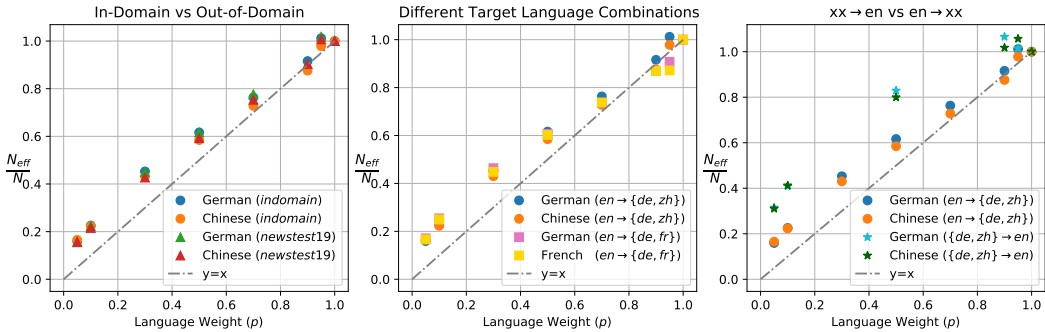

Figure 6: The effective fraction of parameters allocated to each task as estimated by our joint scaling laws. *Left:* Comparison of the capacity splitting behavior of En→{De, Zh} models for in-domain and out-of-domain test sets. We observe minimal differences between the two setups. *Center:* Comparison of the capacity splitting behavior for En→{De, Zh} and En→{De, Fr} models. We don't observe any changes in the interaction between the tasks based on language similarity. *Right:* Comparison of the capacity splitting behavior for translation to and from English. XX→En exhibit more synergy among the tasks.

**Consistency Across Languages Pairs:** In Figure 6 (center), we compare the capacity splitting behavior of En→{De, Zh} and En→{De, Fr} models. The conventional wisdom in the MT literature suggests that the tasks in En→{De, Fr} should exhibit a more positive interaction with each other compared to En→{De, Zh}. This is often justified by the intuition that representations are more aligned in related languages and more aligned representations will encourage parameter sharing (Dabre et al., 2017). Surprisingly, our results suggest that the interaction dynamics in En→{De, Fr} and En→{De, Zh} models are not significantly different. In both settings, we observe a relatively neutral multitask behavior – the performance of MNMT of size $N$ trained on task $i$ with (sampling) weight $p$ is essentially similar to a single-task model of size $pN$. In other words, there is minimal synergy among the tasks in both setups.

**En→XX vs XX→En:** In Figure 6 (right), we compare the interaction between the tasks when translating out of English vs when translating to English. In stark contrast to the En→XX setting, when translating into English, we observe significant positive synergy among the tasks. This observation aligns well with recent results in the literature showing multilingual models achieving SOTA performance for translation to English (Chowdhery et al., 2022; Lepikhin et al., 2020). It is unclear if this synergy arises as a specificity of having English is the target language or because multi-task encoding is intrinsically more amenable to parameter sharing than multi-task decoding. Understanding the exact dynamics giving rise to such positive interaction between the task is an exciting open question.

### 3.4 GUIDING TASK BALANCING

As discussed in the introduction, one of the areas where multitask (multilingual) scaling laws can be most impactful is in guiding task balancing/weighting when training large multitask models, an open problem that has been studied extensively (Aharoni et al., 2019; Wang et al., 2020). However, in its current form, our (joint) scaling law can only be use to decide between weightings that were for used for fitting it and cannot be used to predict performance on new, unseen weightings, as $\beta_{p,i}$ needs to be estimated empirically.

To extend to unseen task weightings, we instead focus on estimating $f_i(\cdot)$. Given access to $f_i(p)$, accurate prediction of $\mathcal{L}_i(N)$ for **any weighting** can be achieved by using the **single-task scaling law**:

$$\mathcal{L}_i(N; p) = \beta_{1,i}\big(\hat{f}_i(p)N\big)^{-\alpha_i} + L_\infty^{(i)}. \tag{11}$$

As observed in Section 3.3, $f_i(p)$ has a series of desirable properties that makes it easy to estimate: (i) it is invariant to test set and languages, (ii) it is smooth and generally well-behaved. As such, one can achieve an accurate approximation of $f$ with just a few data points.

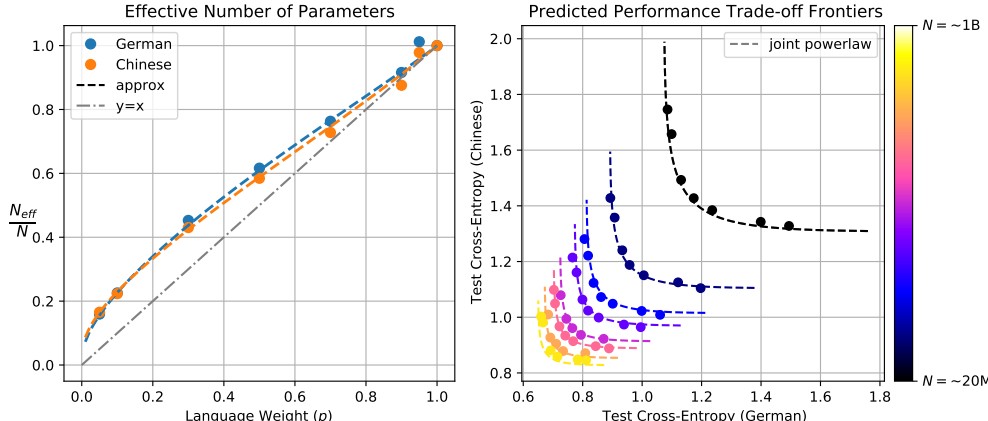

Figure 7: Approximate joint scaling laws described by equations (11) and (12) almost perfectly capture the task interactions across all scales. *Left:* The fitted approximation $\hat{f}$ described in Equation 12. *Right:* The predicted performance trade-off frontier (dashed lines) as well as the empirically observed trade-off values.

We utilize this methodology to estimate the full task performance trade-off frontier for En→{De, Zh} models. For estimating $f_i(\cdot)$, we fit an approximate joint scaling law of the form Equation 11, where $f_i(\cdot)$ is parameterized as

$$\hat{f}_i(p) = p + c_1 p^{c_2} (1 - p)^{c_3} \tag{12}$$

with $c_1, c_2, c_3$ being fitted coefficients. Figure 7 demonstrates our results; our procedure is able to almost perfectly capture the full task performance frontier across a variety of model scales. With access to such accurate predictions of the performance frontier, a practitioner can precisely determine how to weigh the individual tasks during the training based on her preference and target model size.

We should note that the choice of function class to fit $f_i(\cdot)$ is highly dependent on the practitioner's computational budget. In our case, we prioritized accuracy and used a flexible function class of the form (12) for fitting. Such flexibility comes with the cost of needing to compute more empirical values to reliably estimate $f(\cdot)$. In the scenarios with more limited computational budget, we have observed that even rudimentary linear approximations of $f$ are able to provide accurate representations of the performance frontier. See Appendix E for examples.

**Translation Quality**   Finally, we note that in the MT literature, quality is often measured via metrics such as BLEU (Papineni et al., 2002), ChrF (Popović, 2015) and BLEURT (Sellam et al., 2020) as opposed to cross-entropy, since the latter doesn't account for the problem of *decoding* translations from the models and is sometimes found to not correlate with human preferences (Koehn & Knowles, 2017). As such, MT practitioners might be concerned regarding the applicability of these results for practical applications. To ensure that our findings also apply to the quality of translations, we decode translations from our trained models using beam search (Graves, 2012) and evaluate how their quality changes as we scale the models, using ChrF and BLEURT.

Figure 8 (left) shows cross-entropy and ChrF scores for the En→De language pair of our En→{De, Fr} models, evaluated on the in-domain test set. We find that this automatic metric has an almost-linear relationship with cross-entropy, hinting that our observations also generalize from cross-entropy to generation quality. Figure 8 (right) also shows the predicted ChrF performance trade-off frontier obtained by fitting our joint scaling law (Equation 7) to the ChrF performance on the in-domain test set (parametrizing the effective parameter fraction function as in Equation 12). Our procedure is able to capture this trade-off frontier almost as well as the cross-entropy frontier. Similar findings for the BLEURT metric on out-of-distribution test sets can be found in Appendix F.

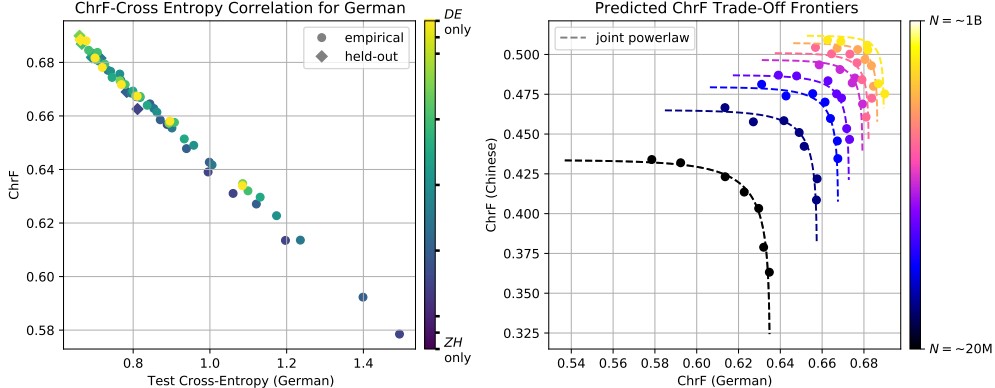

Figure 8: The generation quality behavior of our models as measured by ChrF. *Left:* We observe consistent positive correlations between ChrF and cross-entropy loss. *Right:* Our scaling laws can be used to generate accurate performance trade-off frontiers for ChrF.

## 4 CONCLUSIONS & FUTURE WORK

Current state-of-the-art large neural models are moving towards using as much data from as many domains and modalities as possible to unlock exciting new capabilities. Unfortunately, as of yet, the research community does not have a clear understanding of the behavior of these multitask models at scale. This in turn slows down the model development process since practitioners have to resort to trial and error for balancing their tasks in their models. In this paper, we attempted to take an initial step towards alleviating this problem by performing a large-scale study of the properties of models trained to solve multiple task.

In particular, we attempted to study this problem from the lens of multilingual machine translation. We showed that, for each task and each task weighting, a power-law describes the evolution of the model test performance as a function of the model size. We examined the dependence of the scaling law parameters on the task weights and demonstrated that the scaling exponent and the irreducible loss are independent of the task weightings. Using these observations, we provided a novel joint scaling law that succinctly captures the scaling behavior across different model sizes and task weightings and used it to define the notion of *effective fraction of parameters* assigned to a task ($f_i(\cdot)$). We showed that this quantity robustly captures the task interactions and is surprisingly invariant to the similarity of the tasks. In the end, we sketched a procedure to use $f_i$ to estimate the task performance trade-off frontier for all model scales.

**Future Work** In this paper, we attempted to study the scaling behavior of multitask models. In order to keep our investigation tractable, we focused our study on MNMT models. Examining whether the conclusions of our work apply to setups beyond translation is a promising research direction. In the experiments presented in the paper, we focused only on the two-task scenario. We believe the presented results should be easily extendable to the multitask setup. We leave this to future work. Finally, to simplify the model scaling behavior, we focused our analysis to the data rich setup. However, in many applications, at least some of the tasks are mid- or low-resource. Extending these results to such scenarios is an interesting future direction.

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

# A  MODEL SIZES AND HYPERPARAMETERS

| Enc. Layers | Dec. Layers | Emb. Dim | # Heads | Head Dim | MLP dim | Vocab Size | # Parameters | Corrected # Parameters |
|---|---|---|---|---|---|---|---|---|
| 2 | 2 | 512 | 8 | 64 | 2048 | 128k | 149,953,024 | 18,881,024 |
| 3 | 3 | 768 | 12 | 64 | 3072 | 128k | 260,322,816 | 63,714,816 |
| 6 | 6 | 768 | 12 | 64 | 3072 | 128k | 324,035,328 | 127,427,328 |
| 9 | 9 | 768 | 12 | 64 | 3072 | 128k | 387,747,840 | 191,139,840 |
| 9 | 9 | 1024 | 16 | 64 | 4096 | 128k | 601,931,776 | 339,787,776 |
| 12 | 12 | 1024 | 16 | 64 | 4096 | 128k | 715,193,344 | 453,049,344 |
| 12 | 12 | 1280 | 16 | 80 | 5120 | 128k | 1,035,876,864 | 707,869,184 |
| 12 | 12 | 1536 | 16 | 96 | 6144 | 128k | 1,412,528,128 | 1,019,312,128 |

# B  INDIVIDUAL SCALING LAWS FITS

## B.1  OUT-OF-DOMAIN

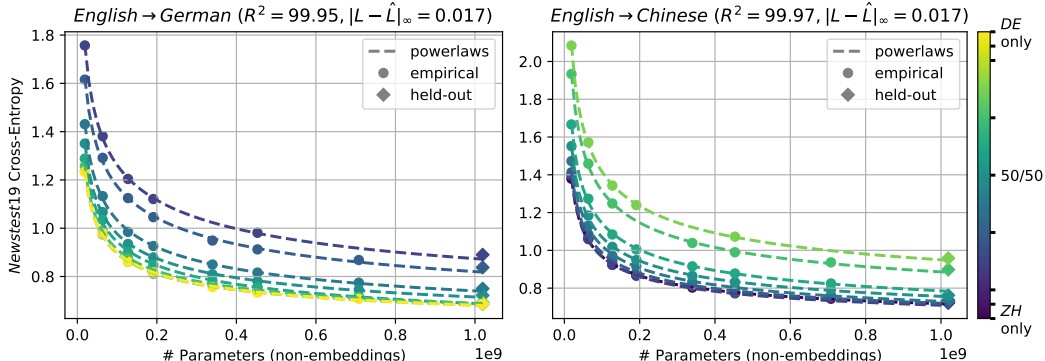

Figure 9: The evolution with model size of the cross-entropy loss on the *newstest19* test set for En→{De, Fr} models, as well as the fitted scaling laws. The color represents the weighting of the languages. Note that we don't show the *zero-shot* behavior.

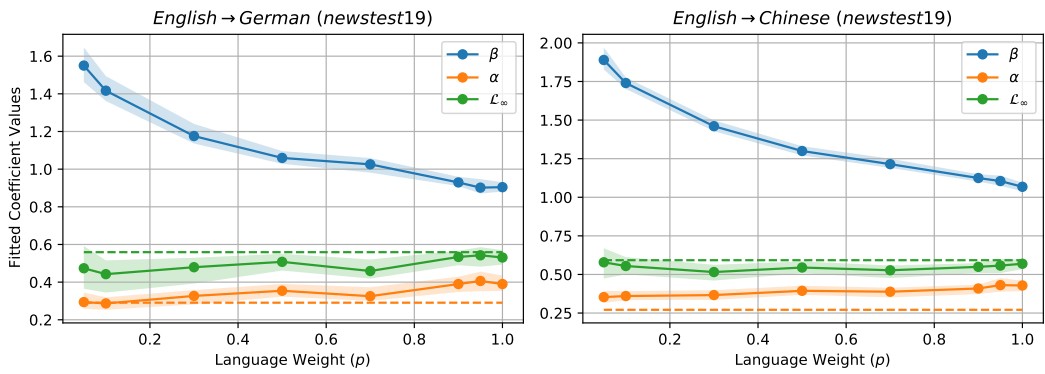

Figure 10: Coefficient values, for scaling laws fitted on *newstest2019*, for German (left) and French (right) as a function of the language weight, with the shaded region representing the standard deviation. The dashed lines represent the value of jointly fitted coefficients from Equation 7

## B.2  ENGLISH→GERMAN, FRENCH

## B.3  GERMAN, CHINESE→ENGLISH

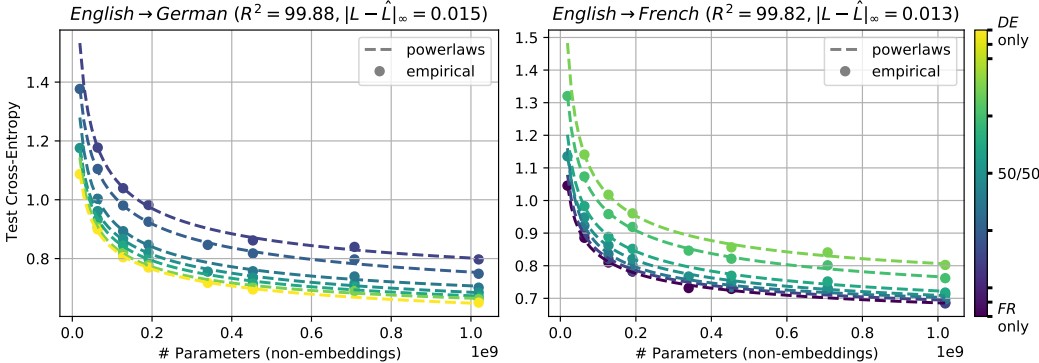

Figure 11: The evolution of the (in-domain) test cross-entropy loss with model size for En→{De, Fr} models, as well as the fitted scaling laws. The color represents the weighting of the languages. Note that we don't show the *zero-shot* behavior.

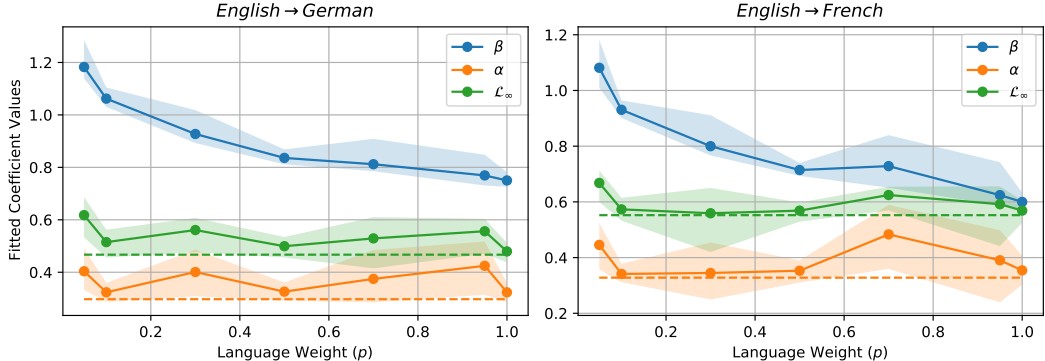

Figure 12: Coefficient values for German (left) and French (right) as a function of the language weight, with the shaded region representing the standard deviation. The dashed lines represent the value of jointly fitted coefficients from Equation 7

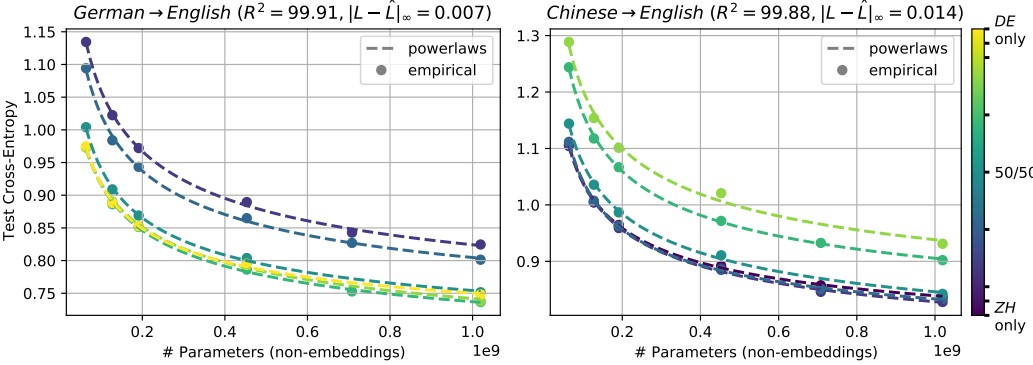

Figure 13: The evolution of the (in-domain) test cross-entropy loss with model size for {De, Zh}→En models, as well as the fitted scaling laws. The color represents the weighting of the languages. Note that we don't show the *zero-shot* behavior.

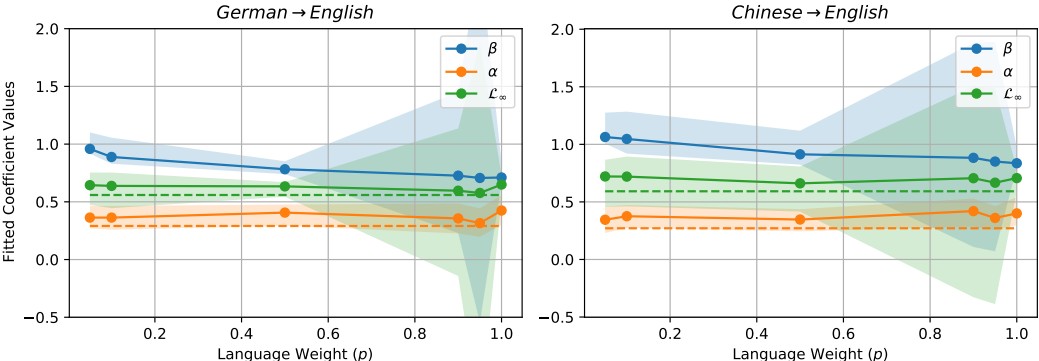

Figure 14: Coefficient values for German (left) and French (right) into English as a function of the language weight, with the shaded region representing the standard deviation. The dashed lines represent the value of jointly fitted coefficients from Equation 7

# C JOINT SCALING LAW FITS

## C.1 OUT-OF-DOMAIN

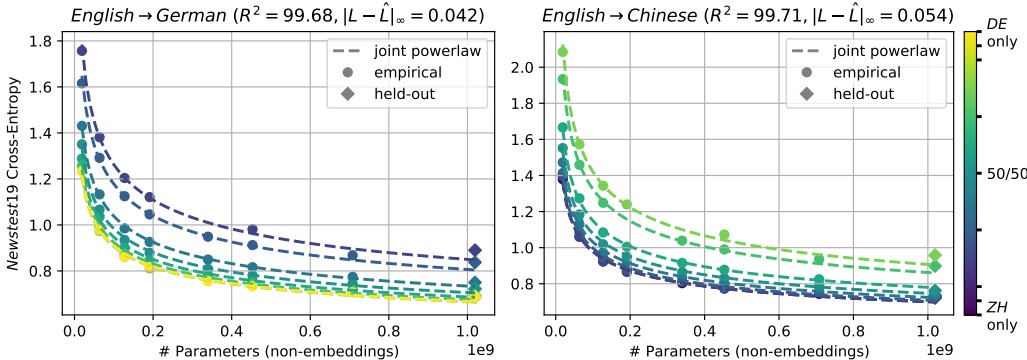

Figure 15: The **joint** scaling law (Equation 7) fitted to models trained for En→{De, Zh} models. Test loss here is evaluated on the *newstest2019* test set.

## C.2 ENGLISH→{GERMAN, FRENCH}

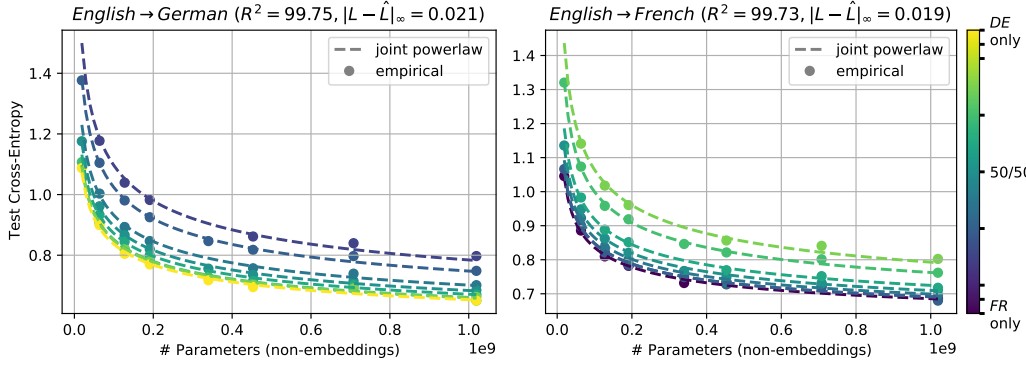

Figure 16: The **joint** scaling law (Equation 7) fitted to models trained for En→{De, Fr} models. Test loss here is evaluated on in-domain test sets.

## C.3 {GERMAN, CHINESE}→ENGLISH

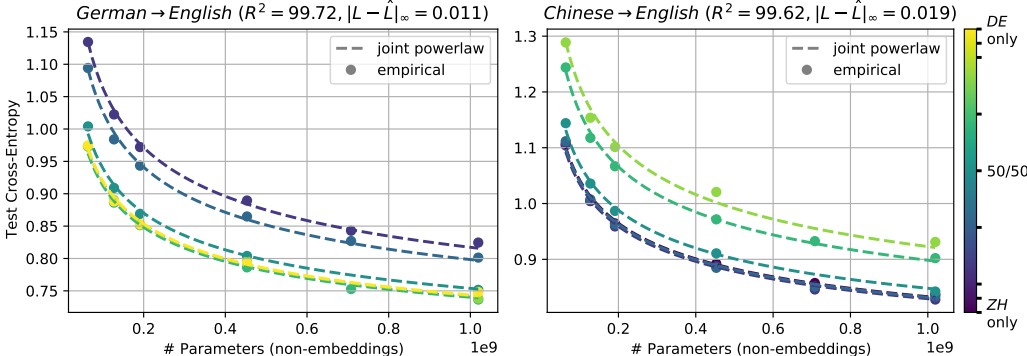

Figure 17: The **joint** scaling law (Equation 7) fitted to models trained for {De, Zh}→En models. Test loss here is evaluated on in-domain test sets.

# D   DERIVATION OF THE EFFECTIVE NUMBER OF PARAMETERS

$$
\begin{aligned}
\mathcal{L}_i(N; p) &= \beta_{p,i} N^{-\alpha_i} + L_\infty^{(i)} \\
&= \beta_{1,i} \left( \frac{\beta_{p,i}}{\beta_{1,i}} \right) N^{-\alpha_i} + L_\infty^{(i)} \\
&= \beta_{1,i} \left( \left( \frac{\beta_{p,i}}{\beta_{1,i}} \right)^{-\frac{1}{\alpha_i}} N \right)^{-\alpha_i} + L_\infty^{(i)} \\
&= \beta_{1,i} \left( \left( \frac{\beta_{1,i}}{\beta_{p,i}} \right)^{\frac{1}{\alpha_i}} N \right)^{-\alpha_i} + L_\infty^{(i)} \\
&= \beta_{1,i} N_{\text{eff}}^{-\alpha_i} + L_\infty^{(i)} \\
&= \mathcal{L}_i(N_{\text{eff}}; p)
\end{aligned}
$$

# E   OTHER APPROXIMATIONS TO THE EFFECTIVE PARAMETER RATIO

We use a linear approximation of the form

$$
\hat{f}_i(p) = c_1(p - 1) + 1. \tag{13}
$$

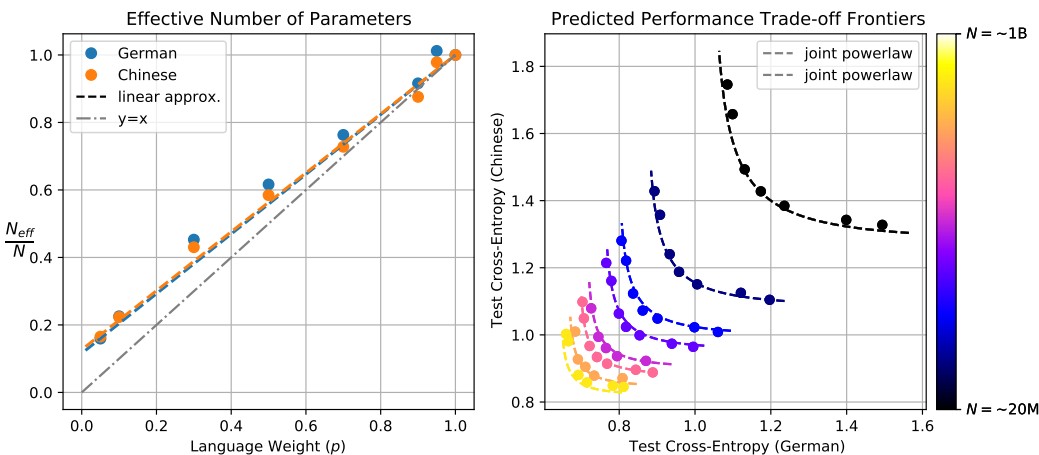

Figure 18:  Approximate joint scaling laws described by equations (11) and (13) is able to capture the task interactions across all scales well, even with single fitted coefficient for ratio function. *Left:* The fitted approximation $\hat{f}$ described in Equation 12. *Right:* The predicted performance trade-off frontier (dashed lines) as well as the empirically observed trade-off values.

# F   TRANSLATION QUALITY

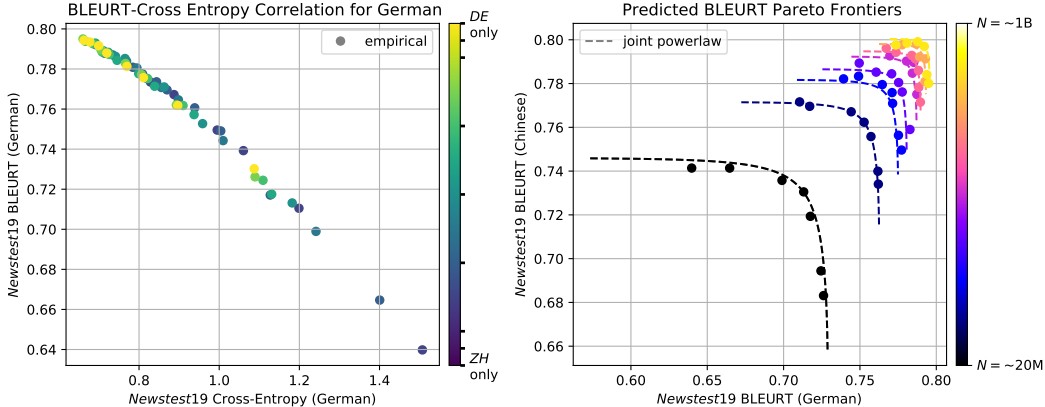

Figure 19: (left) shows cross-entropy and BLEURT scores for the En→De language pair of our En→{De, Fr} models, evaluated on the *newstest19* test set. We find that this automatic metric has an almost-linear relationship with cross-entropy, hinting that our observations also generalize from cross-entropy to generation quality. Figure 8 (right) also shows the predicted BLEURT performance trade-off frontier obtained by fitting our joint scaling law (Equation 7) to the BLEURT performance on the *newstest19* test set (parametrizing the effective parameter fraction function as in Equation 12).

## G  CONVERGENCE CORRECTION

Due to *implicit* scalarization, models trained with very little task weight ($< 0.1$) will see less than a full epoch of that task's data, even when trained with 1M steps. I our experiments we saw that this was causing problems in the fit the scaling laws due to an *undertraining* of our largest models.

To mitigate this problem without training these models for a prohibitively large number of steps, we apply recent findings in learning curve (Hutter, 2021) to estimate the performance of largest models trained with $p \leq 0.05$ task weight at convergence, by fitting a power-law to the performance evolution as training progresses, and predicting the performance of these models at 2.5M steps. This only affect two models per scenario considered.

