# OpenReview forum: "Understanding Multi-Task Scaling in Machine Translation"
_ICLR.cc/2023/Conference — Submitted to ICLR 2023_

### Official Review · Reviewer_JKLj · 2022-10-21

**Confidence:** 3
**Correctness:** 4
**Technical Novelty And Significance:** 3
**Empirical Novelty And Significance:** 4
**Recommendation:** 8

**Clarity, Quality, Novelty And Reproducibility:**

The paper is overall clear and readable. Some important terms such as "capacity splitting behavior" are not really introduced of referenced, but the meaning is quite straightforward. The performance trade-off curves discussed in 3.4 (p.8) are never properly defined--Although the notion is fairly intuitive, I would recommend defining the concept properly instead of wasting space on Fig. 1, which is neither described nor referenced.

The paper is well organized, covers the claimed observations progressively and convincingly, and does a good job conveying its many empirical findings. Those offer novel insights into MNMT training, as far as I am aware.

Strict reproducibility of the paper's experiments seems difficult because most of the data is not available and the exact constitution of the model is not clear. However, the idea of the paper is clear and it should be straightforward for someone "skilled in the art" to replicate these experiments on their data. In fact, given the limitations acknowledged in the "Future work", it would be useful to have the authors of others see how this generalizes to different contexts.

Misc:
(p.2) f_i(w) is used in the last observation before Sec. 2, but never introduced before. (could be introduced in the second observation)
"Figure 4 curves" (14 lines from bottom of p.5) are likely Figure 2 curves.
(p.7) "English is the target language" -> English as the target language

**Strength And Weaknesses:**

This is an enjoyable paper addressing the important issue of how MNMT systems learning operates. The paper is clearly written, there obviously is a massive amount of experimental work behind these results, which show impressive agreement with the theoretical models adopted. I learned a significant amount by reading this paper.

The main obvious weaknesses are actually addressed in the "Future work". For example, given the claimed purpose of the paper, it would have been better to see more than two "tasks". There are also some unresolved questions, for example whether the distinct behaviour of XX->En vs. En->XX models (Sec. 3.3, last par.) is due to the choice of English (a particularly simple language) or reflects some intrinsic multitask learning feature.

**Summary Of The Paper:**

The paper studies scaling laws of multitask neural models in the context of multilingual neural machine translation (MNMT). Scaling is studied in the context of model parameter scaling, in the large data and computational resources condition. The authors show empirically that the scaling law derived from prior work is quite accurate in their context, and that some parameters are invariant to task weighting, while the scaling factor (beta) is not. They introduce the notion of effective number of parameters, which turns out to be useful to compare various training setups and again show some invariance of the scaling behaviour w.r.t. domain or task/target language combination. This also provides a way to address the issue of how to balance the tasks/languages during learning in order to reach the best overall performance.

**Summary Of The Review:**

The paper studies a simple but effective model of how neural models learn multilingual machine translation, offering several non-trivial findings along the way. This will be useful to researchers looking for the best way to train MNMT models, and may have further implications for multitask learning in general.

---

> ### Author Response · Authors · 2022-11-16
> **Response**
>
> | “For example, given the claimed purpose of the paper, it would have been better to see more than two "tasks".
>
> We agree that analyzing the setting where models are trained with more than two tasks is important. As an early effort to understand if our findings apply to more than two tasks, we trained various model sizes for to translate into *three* languages (EN->{DE,ZH,FR}), and compared the predictions using the scaling laws for models trained on two language-pairs (EN->{DE,ZH} and EN->{DE,FR}).
>
> https://imgur.com/a/X5wOEwZ
>
> Overall we find that (combination of) the joint scaling laws fitted on models trained on two language pairs predict well the performance of models trained for three language pairs, showing that the invariances found in previous sections generalize to settings with more than two tasks. These results also hint that computation of effective parameters counts for multi-task models with many tasks can be simplified and made more tractable by training models with much smaller subset of tasks.
>
> We also show how the performance in german evolves as we change the probabilities in the other languages: https://imgur.com/a/rojVGhh
>
> | “There are also some unresolved questions, for example whether the distinct behavior of XX->En vs. En->XX models (Sec. 3.3, last par.) is due to the choice of English (a particularly simple language) or reflects some intrinsic multitask learning feature.”
>
> We agree that there are indeed some interesting questions regarding XX->En vs. En->XX. We believe that there is indeed a distinct behaviour caused by the differences between encoding vs decoding that are not only attributable to English as the target language, but leave a more extensive exploration of this question (by training multi-source models to languages other than English) to future work,

---

### Official Review · Reviewer_3K3H · 2022-10-24

**Confidence:** 4
**Correctness:** 3
**Technical Novelty And Significance:** 3
**Empirical Novelty And Significance:** 3
**Recommendation:** 6

**Clarity, Quality, Novelty And Reproducibility:**

- The paper is with clear writing. There should be more elaborations on the observations though.
- Extra question - I am interesting in this technical question if possible: with a batch size of 500K tokens, do you see any improvement by training the model with 500K gradient steps? (e.g. compared to training the model with 70K steps with the same batch size, for instance).

**Strength And Weaknesses:**

- Strength

This paper is with interesting findings and solid experiment results.

- Weakness

While the paper contributes interesting findings, I found found the explanation and the practical terms of these findings are not that well-explained. For instance, the authors show that there is a relatively little change of some fitted coefficients of the scaling laws, but why does it happen and what possibly it means in practice term are discussed in a too short way. Another example is Equation 10. I am not sure why Equation 10 means the fraction of parameters is independent of the model size. I hope the authors would elaborate more these things in the final version of the paper. I am happy to raise my scores to 8 if these weakness are addressed.

**Summary Of The Paper:**

This paper showed several interesting findings:
1. showed that the scaling laws fits well to the empirical test-cross entropy performance of a model at the end of the training.
2. showed that not all the fitted coefficients of the scaling laws vary that much when we change the sampling rate.  Specifically, alpha and L seem to be relatively constant regardless of the sampling rate. I found this very hard to understand why it happens but it is a very interesting finding.
3. based on the finding, the authors proposed a joint modeling scaling multitask scaling that has (much) less number of parameters (i.e. alpha and L are independent of the task weights) that still captures well the scaling behavior.
4. studied the effective number of parameters with the assumption that the fitted coefficients of the scaling laws are task-independent, showing that the fraction of parameters allocated to task only depends on beta and alpha (beta is sampling-rate specific while alpha is not).
5. sketched a procedure that estimates the task performance trade-off frontier for all model scales

**Summary Of The Review:**

A good paper, recommend to be accepted.

---

> ### Author Response · Authors · 2022-11-11
> **Response**
>
> |  “While the paper contributes interesting findings, I found found the explanation and the practical terms of these findings are not that well-explained. For instance, the authors show that there is a relatively little change of some fitted coefficients of the scaling laws, but why does it happen and what possibly it means in practice term are discussed in a too short way”
>
> We thank the reviewer for their comments. We fully agree that the empirical observations regarding constant scaling coefficients merit a more expansive discussion. We will further extend the discussion in Section 3.2 as follows:
>
> “(...) As the figure suggests, the lines are all near parallel, suggesting that the scaling exponent is unchanged for all $p$.
>
> This invariance in the estimates of $\\mathcal{L}_{\infty}$ and $\alpha$ has important implications: First, the fact that irreducible loss is independent of the sampling probability suggests that multi-task training does not significantly change the information content of the data for any of the tasks in the model. Given that we operate in the data-rich regime, this observation is not too surprising. Nevertheless, this is a good sanity check to have. Moreover, the invariance of the scaling exponents to the sampling probability suggests that, for high-resource tasks, simply scaling the model size is enough for delivering predictable gains across all the tasks in the model. As such, attempting to achieve a perfect balance between the different tasks in the model might not be necessary at scale. As a concrete example, for our translation tasks, we observe the (reducible) test loss improves by a factor of $\approx 0.81$ with each doubling of the model size, regardless of the training data mixture. Finally, given that even single-task models follow this very same trend, this also means that single-task scaling laws can be used to gauge the benefits of scaling multitask models."
>
> |  “Another example is Equation 10. I am not sure why Equation 10 means the fraction of parameters is independent of the model size.
>
> While the total number of effective parameters (Eq. 9) does indeed depend on the model size (since $N$ appears on the right-hand side), the **fraction** of effective parameters (Eq. 10) for a given task weighing $p$ is solely dependent on the scaling exponent $\alpha$ and scaling multipliers $\beta_p$ and $\beta_1$. Since these coefficients are constant for a given combination of tasks and independent of the model size, the fraction of effective parameters is also independent of model size and depends solely on the nature of the tasks and their interaction.
> This means that if, for example, a mNMT model with 10M parameters has 60% effective parameters (6M) allocated to a particular language pair, then a model with 1B parameters trained for the same language-pairs/data will also have same 60% effective parameters (600M) associated with that same language pair.
>
> We agree that this observation might not be clear from the current text. We will further elaborate on this point in an updated version of the paper.
>
> |  “Extra question - I am interesting in this technical question if possible: with a batch size of 500K tokens, do you see any improvement by training the model with 500K gradient steps? (e.g. compared to training the model with 70K steps with the same batch size, for instance).”
>
> We found that this is highly dependent on model size. For example, for our smallest models (2-3 layers), there wouldn’t be much of a difference between training for 70k steps and 500k steps. However bigger models require more steps for convergence, with 9L models requiring almost the 500k steps. This was the reason for increasing the number of training steps for larger models to 1M steps.
>
> For reference, here is the cross-entropy evolution on both English->German and English->Chinese for a base (12L) model trained with 50% weight on each language-pair, where we can see that performance keeps improving way beyond the 70k steps.
>
> https://imgur.com/a/uRhxPCf

---

### Official Review · Reviewer_49op · 2022-10-24

**Confidence:** 4
**Correctness:** 3
**Technical Novelty And Significance:** 2
**Empirical Novelty And Significance:** 3
**Recommendation:** 5

**Clarity, Quality, Novelty And Reproducibility:**

All the clarity, quality, novelty, and reproducibility strengths and weakenesses are detailed in the section Strength And Weaknesses.

**Strength And Weaknesses:**

Strengths:
- the paper is mostly clear and well-written. I had no issue to undetstand it.
- the method and experiments are reproducible. I am confident that I would be able to reproduce the shape of the curves but may be not with similar points since the evaluation framework used is not detailed enough.
- the paper delivers a comprehensive analysis of the scaling of multilingual NMT.

Weaknesses:
- I find very confusing that in this work multingual NMT is multi-task. NMT, multilingual or not, has only one task: translate. A "task" in this paper is a translation direction. In other word, the paper isn't investigating the multi-task, but multilingual, scaling laws. This is a critical issue I think since I have no idea how the conclusions/observations from the paper could be applied to a truly multi-task model.
- The paper isn't motivated enough. Why scaling laws for bilingual NMT are expected to be different from multilingual NMT? What are the challenges? What are the research questions to be addressed here?
- The contributions over previous work are unclear to me. The methodology from previous work on the scaling laws of bilingual NMT is applied to multilingual NMT. Experiments are performed and analyzed, but I'm unclear how this work changes the methodology (does it?). Maybe a a more extended section on the scaling laws of bilingual NMT, highlighting the limits of previous studies addressed in this paper, would be necessary to help the reader to spot the contributions.
- The limit of this work are not clearly discussed.

**Summary Of The Paper:**

From the assumption that more and more models are proposed and evaluated in a multi-task setting, this paper proposes to explore the multi-task scaling laws, with a focus on multilingual machine translation. This focus is motivated by the abundance of benchmarks for this task and the existence on previous work about the scaling laws of standard bilingual NMT systems.

They trained 200 NMT systems.
Among the contributions:
- they propose a method to determine the number of parameters allocated to each translation direction in the model.
- they provide a method to predict the full task performance given the model size
- they found that the relationship between languages doesn't impact much the scaling behavior of the model




**Summary Of The Review:**

The paper needs better motivation and should fully assume that it is about multilinguality rather than multi-tasking.  I strongly disagree that multilingual models are multi-task models.
The analysis and observations made from the experiments are very interesting and would be insightful for future work on multilingual NMT. On the other hand, it is unclear how different is the methology used by previous work on scaling laws and whether it needed special adaptations to the multilingual scenario.

---

> ### Author Response · Authors · 2022-11-11
> **Response**
>
> We thank the reviewer for their comments.
>
> |  “I find very confusing that in this work multingual NMT is multi-task. NMT, multilingual or not, has only one task: translate. A "task" in this paper is a translation direction. In other word, the paper isn't investigating the multi-task, but multilingual, scaling laws. This is a critical issue I think since I have no idea how the conclusions/observations from the paper could be applied to a truly multi-task model.”
>
> Multilingual NMT has been framed as a multi-task optimization problem extensively in the past [1,2,3,4,5]. This stems from the fact that both learning to translate into/from two (or more) languages and learning two (or more) "real tasks" (e.g. translation and summarization) can both be seen as learning two (or more) different functions, and there is little difference in the mathematical formulation of the optimization problem.
> However, we fully agree that there is a need to examine the scaling behavior of "truly" multi-task models. We set up this study as a first step in this path and we aim to explore more diverse setups in our future work.
>
> [1] Luong et. al, 2015, Multi-task Sequence to Sequence Learning
>
> [2] Firat et. al, 2016, Multi-Way, Multilingual Neural Machine Translation with a Shared Attention Mechanism
>
> [3] Arivazhagan et. al, 2019, Massively Multilingual Neural Machine Translation in the Wild: Findings and Challenges
>
> [4] Jean et. al, 2019, Adaptive Scheduling for Multi-Task Learning
>
> [5] Wang et. al, 2020, Gradient Vaccine: Investigating and Improving Multi-task Optimization in Massively Multilingual Models
>
> |  “The paper isn't motivated enough. Why scaling laws for bilingual NMT are expected to be different from multilingual NMT? What are the challenges? What are the research questions to be addressed here?”
>
> Previous work around scaling laws for bilingual MT (or any other setting to our knowledge) studies how scale affects performance when we consider training/test data as single task, and if we considered the mNMT problem as single objective (say, by looking at the average cross-entropy on the different language pairs) we expect to see similar findings to previous work on scaling laws for bilingual MT. The key difference in our work comes from considering the training data not as a monolithic, single-task distribution with a single learning objective, but as the combination of several different tasks/language-pairs, and the learning problem as multi-objective optimization.  This comes with the added challenge of understanding how the weight of the different objectives (language pairs) affects the performance on the individual tasks/language-pairs, which we do by exploring different weightings in section 3.2. This formulation also allows us to study how task similarity affects the scaling on the individual, something that wasn't possible with previous works.
>
> |  “The contributions over previous work are unclear to me. The methodology from previous work on the scaling laws of bilingual NMT is applied to multilingual NMT. Experiments are performed and analyzed, but I'm unclear how this work changes the methodology (does it?). Maybe a a more extended section on the scaling laws of bilingual NMT, highlighting the limits of previous studies addressed in this paper, would be necessary to help the reader to spot the contributions.”
>
> As mentioned, previous studies around scaling laws don’t consider the training objective as a multi-task optimization problem. This means that all the methodology surrounding the impact of the tasks’ weight on the loss is novel, including (1) the analysis of the impact of the task’s weight on scaling coefficients, (2) the proposed joint scaling law, (3) the analysis of the effective parameters for each tasks in a multi-task model and (4) the prediction of the full tasks’ performance trade-off frontier.
> However, while we did mention that short-comings of previous work’s analysis in the introduction, we agree that the differences from previous work are not highlighted enough. We will include a more detailed description of these differences in the final version.

---

### Official Review · Reviewer_TBUV · 2022-10-28

**Confidence:** 4
**Clarity, Quality, Novelty And Reproducibility:** Please see the previous section.
**Correctness:** 3
**Technical Novelty And Significance:** 2
**Empirical Novelty And Significance:** 2
**Recommendation:** 5

**Strength And Weaknesses:**

Strengths:
1.	The purpose of this work is to finally gain a clear understanding of the behavior of these large-scale multi-task models by performing a large-scale study of the properties of models trained to solve multiple tasks.
2.	This work provides a comprehensive analysis of the proposed methodology, allowing the reader to gain a detailed understanding of the work.

Weaknesses:
(1) The designed experiments are not comprehensive enough and need to cover more scenarios.
(2) The purpose and details of the experimental design need to be described more clearly.

Questions:
1.	For the effective number of parameters, do different translation languages have overlapping parameters? Is it possible to provide a demonstration of the parametric distribution in experiments to judge the effectiveness of the effective number of parameters?
2.	For translation quality experiments, using ChrF and BLUERT are conventional choices. I suggest that the effects of settings of different values be shown as a case study. Through different cases, MT literature may show different characteristics, which can inspire future research.
3.	For Equation 12, how are c1, c2, and c3 obtained?  What effect do these three parameters have on fi?
4.	For the experiment, I suggest that a set of experiments can be added. There was no correlation between the groups of languages in the translation task in this experiment. Its purpose is to test the robustness of the proposed method.


**Summary Of The Paper:**

This work provides a large-scale empirical study of the scaling properties of multitask/multilingual neural machine translation models. The work examined the dependence of the scaling law parameters on the task weights and demonstrated that the scaling exponent and the irreducible loss are independent of the task weightings. Based on the above observations and analysis, the paper provided a novel joint scaling law that captures the scaling behavior across different model sizes and task weightings and used it to define the notion of the effective fraction of parameters assigned to a task. Through experiments, this quantity captures task interactions robustly and is surprisingly invariant to task similarity. Finally, the paper draft a procedure that uses fi to estimate task performance trade-off bounds for all model scales.

**Summary Of The Review:**

Please see the previous section.

---

> ### Author Response · Authors · 2022-11-11
> **Response and Clarification Questions**
>
> We thank the reviewer for their comments.
>
> | “The designed experiments are not comprehensive enough and need to cover more scenarios.”
>
> What exact extra experiments would you like to see? It was unclear what these would cover and what questions they would help answer.
>
> | “For the effective number of parameters, do different translation languages have overlapping parameters?”
>
> The current formulation only allows calculating the effective number of parameters wrt a single-task model trained for a task/language-pair. In cases where both languages-pairs have high effective parameter ratio (say 75%/75% ratios for a model trained with 50%/50%) it’s natural to infer that this model must be sharing parameters between both tasks, but the scaling laws don’t allow predicting exactly what parameters are being shared.
>
> | “Is it possible to provide a demonstration of the parametric distribution in experiments to judge the effectiveness of the effective number of parameters?”
>
> Would it be possible to clarify what this distribution would be over? And how it would help measuring the “effectiveness” of the effective number of parameters?
>
> | “For translation quality experiments, using ChrF and BLUERT are conventional choices. I suggest that the effects of settings of different values be shown as a case study. “
>
> Is it possible to clarify what values we should vary and study for this case study?
>
> | “For Equation 12, how are c1, c2, and c3 obtained? What effect do these three parameters have on fi?”
>
> These parameters are fitted automatically along with the other parameters of the scaling law ($\alpha$, $\beta$, and $\mathcal{L}_\infty$). This functional form was inspired by the incomplete beta function, and the effects of these parameters are similar to the ones in the CDF of the beta distribution (see https://en.wikipedia.org/wiki/File:Beta_distribution_cdf.svg)
>
> | “For the experiment, I suggest that a set of experiments can be added. There was no correlation between the groups of languages in the translation task in this experiment. Its purpose is to test the robustness of the proposed method.”
>
> To clarify, would these extra experiments be with different language pairs? If so, we agree that they would certainly be useful, but given that we explored three different language-pair-combination setups, each with different properties (different languages, similar languages, and reverse direction), and the computation cost of each full setup, we leave the exploration of other setups to future work.

---

### Decision · Program_Chairs · 2023-01-20

**Decision:**

Reject

**Justification For Why Not Higher Score:**

There is little in the way of novel and useful knowledge in the paper. The discussion of "multitask" methods is confusing and negatively impacts the validity of the results. Finally, the paper does not go into more depth into some of the unresolved mysteries of multilingual MT such as is there something special about English.

**Justification For Why Not Lower Score:**

N/A

**Metareview: Summary, Strengths And Weaknesses:**

This paper describes a large-scale empirical study of scaling in multilingual machine translation. In particular, the paper studies the relation to model size and language direction weights on model performance. The paper finds out that language similarity has little effect on performance, but the system is affected by what side of the translation English stands on.

Most of the findings of the paper are not terribly surprising or novel, but it is nice to have them confirmed in a systematic study. The paper may help researchers who are training multilingual NMT systems.

The paper attempts to generalize from multilingual NMT to multitask training, but that's a reach. A more relevant generalization is a multilingual system for a particular task. The inclusion of "multitask" is confusing. The paper also fails to make any deeper hypothesis and conclusions -- for example is English special and the results of it being on different sides of the translation relevant only to English language pairs. Lastly, most of the observations of the paper are already established in the field.